# Parameter Optimization for Printing Barium Titanate Piezoelectric Ceramics through Digital Light Processing

**DOI:** 10.3390/mi14061146

**Published:** 2023-05-29

**Authors:** Dongcai Zhang, Yaodong Yang, Wei-Feng Rao

**Affiliations:** Faculty of Mechanical Engineering, Shandong Institute of Mechanical Design and Research, Qilu University of Technology (Shandong Academy of Sciences), Jinan 250353, China; 10431200111@stu.qlu.edu.cn

**Keywords:** additive manufacturing, digital light processing, barium titanate, piezoelectricity

## Abstract

Digital light processing (DLP) technology has emerged as a promising 3D printing technology with the potential for the efficient manufacturing of complex ceramic devices. However, the quality of printed products is highly dependent on various process parameters, including slurry formulation, heat treatment process, and poling process. This paper optimizes the printing process with respect to these key parameters, such as using a ceramic slurry with 75 wt% powder content. The employed degreasing heating rate is 4 °C/min, the carbon-removing heating rate is 4 °C/min, and the sintering heating rate is 2 °C/min for heat treatment of the printed green body. The resulting parts are polarized using a poling field of 10 kV/cm, a poling time of 50 min, and a poling temperature of 60 °C, which yields a piezoelectric device with a high piezoelectric constant of 211 pC/N. To demonstrate the practical application of the device, its use as a force sensor and magnetic sensor is validated.

## 1. Introduction

Piezoelectric ceramics are functional materials that can achieve force–electric conversion through the piezoelectric effect and are widely used in multiple fields such as biology [1,2], military [3,4,5], and renewable energy [6]. Lead-free piezoelectric ceramics have become the trend of future piezoelectric material development due to their environmental friendliness and safety [7]. Among them, barium titanate (BTO) material has become the most widely used lead-free piezoelectric material due to its simple preparation process and clear theoretical basis [8]. Currently, the manufacture of lead-free piezoelectric ceramic devices with complex shapes is still a significant issue.

Additive manufacturing (AM) technology is considered to be the best process for forming complex structural devices because of its unique two-dimensional stacking molding method. The existing AM technologies capable of achieving lead-free piezoelectric ceramic forming include stereolithography (SLA) [9,10,11], extrusion freeform fabrication (EFF) [12], fused deposition modeling (FDM) [13,14,15,16], direct ink writing (DIW) [17], and digital light processing (DLP) [18,19,20]. DLP molding technology has become the first choice for piezoelectric ceramic printing due to its high efficiency and good performance in piezoelectricity, printing resolution, surface quality, and densification [21,22,23,24]. However, the performance of the parts printed by DLP is affected by multiple factors, such as the selection of slurry composition, choices of heat treatment, and poling schemes. One of the most important factors affecting the printability of ceramic ink is the powder solid content [25]. Sotov et al. [26] believed that a BTO slurry with 30 vol% powder had better rheological properties and lower shear stress between the scraper and release film. Wang et al. [27] found that a BTO ceramic suspension with 40 vol% solid content showed lower viscosity (232 mPa·s) at a shear rate of 46.5 s^−1^ and was more suitable for DLP printing. Cheng et al. [28] demonstrated that a printed ceramic material had a higher piezoelectric constant (166 pC/N) when the powder-solid content was 80 wt%. For research related to the powder-solid content in the DLP printing process of BTO technology, people mainly focus on its effects on slurry rheological properties and piezoelectric performance, but there is little research on its impact on printing efficiency and accuracy (curing depth), despite the fact that powder-solid content has a significant impact on the curing depth of the slurry. In the process of heat treatment, the heating rate constitutes a crucial parameter [29]. Appropriate heating rates can significantly reduce production time and energy consumption while ensuring optimal ceramic density and shape retention. Jiang et al. [30] recommended a sintering temperature of 1425 °C for 2 h to achieve better piezoelectric performance in the device. Liu et al. [31] conducted heat treatment using a degreasing ramp rate of 0.5 °C/min, a carbon removal ramp rate of 1 °C/min, and a sintering ramp rate of 3 °C/min, the total processing time for the heat treatment was approximately 77 h. Chen et al. [32] utilized multiple ramp rates for degreasing treatment, resulting in a total degreasing processing time of up to 35 h. Despite the extensive investigation into the impact of sintering temperature and atmosphere on heat treatment, there has been limited research conducted on the influence of heating rate. Poling treatment is necessary to achieve good piezoelectric properties in devices following heat treatment [33]. Lu et al. [34] demonstrated that a poling field of 6 V/μm enabled ceramics to avoid high voltage breakdown and improved their piezoelectric properties. However, most studies have concentrated solely on the effect of poling field on the piezoelectric constant and a complete analysis of the poling process from the perspective of poling field, time, and temperature is yet to be undertaken.

This article presents an experimental analysis of the slurry composition, printer settings, heating rate, and poling processes in DLP printing technology. The influence of different powder-solid contents on slurry curing depth in the ceramic slurry was discussed. The impact of different heating rates on device densification was investigated. The study explored the influence of different poling fields, poling times, and poling temperatures on piezoelectric devices to achieve optimal piezoelectric performance. An optimized printing process was used to manufacture a high-porosity piezoelectric device, and its application in force–electric sensing and magneto–electric sensing was extended.

## 2. Material and Methods

### 2.1. Raw Materials

Commercial BTO powders (99.9% metals basis, particle diameters d_50_ = 200 nm) were obtained from Shanghai Macklin Biochemical Co., Ltd. (Shanghai, China). The photosensitive resin (YF-RC2110005) was obtained from Zhejiang Xunshi Technology Co., Ltd. (Shaoxing, China). Ingredients were polyurethane acrylate (PUA), acrylic ester (ACM), diphenyl(2,4,6-trimethylbenzoyl)phosphine oxide (TPO), and 4-Acryloylmorpholine (ACMO). The dispersant is DISPERBYK-111, produced by BYK Additives (Shanghai, China) Co., Ltd. (Shanghai, China).

### 2.2. Slurry Preparation

Commercial-grade barium titanate powder (d_50_ = 200 nm) was added to the ceramic-based photosensitive resin, and 3% of the total mass of barium titanate powder and ceramic-based photosensitive resin was added as a dispersant. All materials were added to the ball mill jar, and zirconia ball mill beads of equal mass to the materials were added and ball milled in a ball mill at 500 r/min for 2 h. The mixed slurry mixture was filtered through a 10-mesh sieve, and the filtered slurry was put into a vacuum-drying oven for defoaming. The final barium titanate ceramic slurry suitable for printing by a light-curing printer was obtained.

### 2.3. DLP Printing Program

The barium titanate piezoelectric device is printed by a light-curing printer that uses ultraviolet light to selectively cure the paste at the bottom of the tank so that the cured layer is bonded to the printing platform. After the printing platform is lifted, the tank rotates while the squeegee in the tank scrapes the paste flat, and then the printing platform is lowered for the next layer of printing action. Pour the barium titanate ceramic paste into the ceramic printer trough, convert the designed model file into an STL file, import it into the slicing software, and set the base curing time, base layer number, layer curing time, and other related parameters. Run the machine and wait for the printing to finish. Before running the machine, the printing platform needs to be leveled to ensure that the print and the platform fit together tightly when printing the substrate. The slicing process of the model also requires special attention and should be done with as little variation as possible in the shape of the two adjacent layers (the main thing should be to avoid a proliferation of print areas). During the operation of the machine, attention should be paid to the adjustment of the squeegee height to ensure that the paste is evenly spread in the print slot. After printing, the print needs to be cleaned initially to facilitate the complete removal of the print from the printing platform. If necessary, a print substrate can be added to the print model to ensure the integrity of the used part. Printing parameters need to be designed, including five layers of the base layer, 35 s of base layer curing time, 20 μm printing layer thickness, and 5 s of the main body curing time.

### 2.4. Green Body Post-Treatment Process

The two-step debonding process (degreasing-carbon removal) can better remove the non-barium titanate powder part of the material. The degreasing process is carried out under a vacuum or argon atmosphere to slow down the decomposition of resin and dispersant materials and to ensure that the decomposition products slowly evaporate from the print without damaging the shape of the printer body. The carbon removal part is carried out under air to expel the remaining carbon after the decomposition of resin and dispersant, and other materials so that only pure barium titanate powder is left in the printed part after debonding.

Since the internal grain electric domains of BTO ceramic parts are disordered after heat treatment, the parts as a whole do not exhibit piezoelectricity. Therefore, a post-processing procedure such as poling is required to make the printed part piezoelectric. Before poling, a conductive silver paste is brushed on the part and dried in the appropriate position according to the working condition of the part. The ceramic part is placed in a constant temperature oil bath, and the printed part is polarized using a high-voltage DC power supply. The poling field, poling time, and oil bath temperature during the poling process need to be further discussed and determined. The samples used in the heat treatment and polarization experiments are cubes with dimensions of 1 cm × 1 cm × 1 cm, and the relevant performance characterization is conducted on parts with this shape.

### 2.5. Characterization and Testing

The DLP printing device (CeraRay TC-Ⅰ) is provided by Zhejiang Xunshi Technology Co., Ltd. with a printing volume of 64 mm × 40 mm × 200 mm and a resolution of 50 μm. The light curing depth is defined as the surface cleaning and thickness measurement of the resulting sample after irradiation of the paste at the same UV light intensity for different durations. The measurement process uses a five-point measurement method to measure the thickness of the irradiated sample, and the thickness of the five-point measurement is averaged as the curing depth at that length of time. Three samples of each solid content slurry were printed and measured with a micrometer. The densification measurement method used in this paper is the Archimedes drainage method, where the density of the sintered printed part is measured with an electronic density balance, and the densification is calculated. Each heat treatment step was performed for three samples, and the densities of the three samples were measured and averaged. Use vernier calipers to measure the diameter of the graph after overcure several times and calculate the average value. Calculate the overcuring rate using the following metric:(1)$$E=\left[{\left(\frac{d_{0}}{d}\right)}^{2}-1\right]\times 100\%$$
where E is the overcuring rate; d_0_ is the diameter of the actual sample circle (mm); d is the diameter of the design sample circle (mm).

The quasi-static d_33_ tester is the ZJ-4AN quasi-static d_33_ meter developed by the Institute of Acoustics of the Chinese Academy of Sciences. Three polarization experiments were performed for each piezoelectric device with polarization conditions, and the piezoelectric constants were measured and averaged. The voltage signal test is to characterize the RMS value of the voltage that the printed part can produce when the part is excited by ultrasound or force. The force excitation is achieved by a crank rocker mechanism built in-house for multiple cycles of standard force release. The voltage signal is captured and stored by a DS1054Z oscilloscope manufactured by RIGOL Technologies Co., Ltd. (Suzhou, China).

## 3. Results and Discussion

### 3.1. Effects of Solid Content on Curing Depth

For BTO ceramic slurry, it can be considered as a suspension of BTO powder in photosensitive resin. The dispersant is only used to evenly suspend the BTO powder in the slurry. The solid content of BTO powder has an important influence on the printing and post-processing of parts. When the solid content of BTO powder is too high, the flowability of the slurry in the tank deteriorates, making it difficult to form a thin film of slurry on the surface of the tank; the curing performance of the slurry also decreases (curing depth becomes lower), which may reduce the molding efficiency of the part or even lead to molding failure. On the contrary, when the solid content of BTO powder is too low, the overcuring rate of the slurry during molding increases due to the reduction of BTO powder; post-processing becomes more difficult, the shrinkage rate of 3D-printed parts after post-processing increases, and the probability of deformation increases. It is preferred to use BTO ceramic slurry with as high solid content as possible while ensuring that the printed parts have sufficient molding accuracy. After multiple attempts at printing, the failure rate significantly increased when the powder-solid content was around 80 wt%. Therefore, in this article, four slurry samples with solid contents of 65 wt%, 70 wt%, 75 wt%, and 80 wt% were set up for experiments with 5 wt% increments.

Figure 1a shows the corresponding curing depths of the four different solid-content ceramic slurries at different curing times. From the figure, it is easy to see that the curing depth of all solid content samples increases with the increase in curing time. At the same curing time, the curing depth of the ceramic slurry decreases with the increase of solid content. To achieve the maximum printing efficiency of parts while ensuring successful printing, achieving 3D-printed ceramic parts with shorter curing time can improve the printing efficiency as much as possible. When the curing depth is about twice the layer thickness of the printed part, the probability of continuous molding of the printed part is highest. Therefore, a solid content of 75 wt% is the optimal powder-solid content for BTO ceramic slurry. Figure 1b shows the curing depth and overcuring rate measurement of BTO ceramic slurry at different curing times. As shown in the black curve of the figure, the curing depth increases with the increase of curing time. However, the increasing rate of curing depth decreases with the further increase in curing time. According to the curing experiment, when the curing time is set to 5 s, the curing depth of the ceramic slurry is 38 μm, which is sufficient to achieve continuity during the printing process. The red curve in the figure shows the correlation between different curing times and the overcuring rate of printed samples (the calculation method for the overcuring rate is shown in Formula (1)). It can be observed from the figure that the overcuring rate of printed samples increases with the increase of curing time. However, the increasing rate first increases and then decreases. When the curing time is set to 5 s, the average overcuring rate of printed samples is 0.2. Although this has a certain impact on the printing accuracy, considering the overall curing depth, a laminated curing time of 5 s can achieve high-efficiency printing while maintaining relatively high precision. The overcure rate experiment is a circular sample printing experiment, which is calculated by comparing the ratio of the difference between the actual circular area and the designed circular area, as shown in Figure 1b. The calculation formula is shown in Formula (1).

### 3.2. Effects of the Heat Treatment Process

This article discusses the heating rate during each stage of heat treatment (degreasing carbon-removal sintering). A high heating rate can lead to a faster thermal decomposition rate of the ceramic green body, resulting in excessive weight loss and an increased probability of layering, deformation, and surface cracking of the debonded parts. Conversely, a low heating rate will lengthen the degreasing time, not only prolonging the manufacturing process but also wasting a lot of energy. Therefore, it is necessary to use the highest possible heating rate while ensuring the high density and structural integrity of the rebound piezoelectric parts.

Figure 2a shows the density experiments of parts after degreasing and carbon removal processes with heating rates of 1 °C/min, 2 °C/min, 4 °C/min, and 8 °C/min for the ceramic green body. It can be seen from the figure that when the heating rate is increased to 8 °C/min, the density of printed parts decreases significantly, and obvious deformation and cracking phenomena occur in the structure. Therefore, the degreasing and carbon removal process of BTO piezoelectric ceramics adopts a heating rate of 4 °C/min. Figure 2b shows the density experiments of parts after sintering processes with heating rates of 0.5 °C/min, 1 °C/min, 2 °C/min, and 4 °C/min for the ceramic green body. It can be seen from the figure that when the heating rate is 4 °C/min, although the density of printed parts has improved compared to the debonding process, there are still obvious deformation, bending, and layering phenomena in the parts’ structure. Therefore, the sintering process of BTO piezoelectric ceramics adopts a heating rate of 2 °C/min. In summary, the heating curves and environmental atmospheres during the debinding and sintering processes are shown in Figure 2c,d. All heat treatment processes use the above heating rates to ensure sufficient density and structural integrity of the printed parts while optimizing manufacturing efficiency.

### 3.3. Effect of Poling Process

Poling field is the most important parameter in the poling process. As the poling field increases, the force that promotes the orientation and arrangement of ferroelectric domains inside the ceramic becomes stronger, making poling more complete. However, when the poling field is too high, piezoelectric ceramics will undergo breakdown due to inherent properties, causing them to fail. In the experimental process, it was found that the porosity of piezoelectric ceramics prepared by DLP printing technology is lower than that of piezoelectric ceramics prepared by the dry pressing method. Therefore, the breakdown field strength of ceramics prepared by DLP printing technology is 2–3 times lower than that of ceramics prepared by the dry pressing method. After multiple experiments, it was found that when the poling field of barium titanate ceramics reaches 15 kV/cm, the probability of piezoelectric ceramic breakdown greatly increases. Therefore, in this study, poling was carried out using 6 kV/cm, 8 kV/cm, 10 kV/cm, 12 kV/cm, and 14 kV/cm poling field to test the piezoelectric constant using a quasi-static d33 tester. In addition, poling temperature is also an important factor that cannot be ignored in the poling process. With the increase in poling temperature, it becomes easier for the ferroelectric domains in the piezoelectric ceramics to rotate, which is beneficial to the poling process. However, piezoelectric ceramics have a Curie point (Tc). Once the Tc is exceeded, the piezoelectric ceramics will lose their piezoelectric properties. Therefore, the poling temperature should not exceed their Curie temperature (the Curie temperature of barium titanate piezoelectric ceramics is 120 °C) when setting the poling temperature. In this study, poling was performed at poling temperatures of 40 °C, 60 °C, and 80 °C, and their piezoelectric constants were tested. Poling time has an important effect on the poling effect. When the poling time is too long, it not only increases the probability of high-pressure breakdown of printed ceramics but also produces energy waste caused by prolonged voltage application. When the poling field is too low, it cannot provide sufficient power for the rotation of ferroelectric domains, affecting the poling effect. Therefore, in this study, poling was carried out for 20 min, 40 min, 50 min, 60 min, and 70 min, respectively, and their piezoelectric constants were tested. Orthogonal experiments of poling field and poling time were performed under three poling temperatures, as shown in Figure 3.

The piezoelectric constants of the piezoelectric components under the three poling temperatures showed a trend of increasing and then decreasing with the increase of the poling field. This is because with the increase of the poling field, although there is no obvious high-voltage breakdown or burning phenomenon on the surface of the piezoelectric ceramics, slight breakdown tendencies have already occurred in the internal structure, resulting in a decrease in the piezoelectric constant of the piezoelectric ceramics during high-voltage poling. Therefore, adopting a poling field of 10 kV/cm during poling can make the piezoelectric ceramics have better piezoelectric performance. Moreover, under the three poling temperatures, the piezoelectric constants of the piezoelectric components still showed a trend of first increasing and then decreasing with the increase of the poling time. With the increase of the poling time, although the piezoelectric ceramic components did not undergo obvious breakdown or short-circuit phenomena, partially burnt traces could be observed on the surface. This is a precursor to piezoelectric component breakdown, indicating a decrease in the piezoelectric constant value. Therefore, adopting a poling time of 50 min during poling can obtain piezoelectric ceramics with higher piezoelectric constant values. Comparing the piezoelectric constants of piezoelectric parts polarized at three different temperatures, it was found that the optimal poling temperature is 60 °C. This is because a poling temperature of 80 °C is closer to the Curie temperature of BTO ceramics, causing the piezoelectric properties in the ceramics to gradually disappear. Meanwhile, a poling temperature of 40 °C does not effectively promote the orientation of piezoelectric domains, resulting in incomplete directional ordering of some piezoelectric domains.

In summary, to obtain piezoelectric ceramic devices with better piezoelectric constants, a poling field of 10 kV/cm, a poling time of 50 min, and a poling temperature of 60 °C should be used during the poling process. The optimized printing process parameters in this paper are summarized in Table 1.

### 3.4. Print Example: Preparation of Sensor Components

Triply periodic minimal surface (TPMS) structure has the advantages of being lightweight, having high specific strength, and excellent energy absorption, which is mainly used as energy-absorbing devices, lightweight structures, and biomedical implants. As a member of the TPMS family, the P-type structure is easier to print. It has a larger range of porosity regulation, which is more conducive to the manufacture of piezoelectric parts with low weight. A high porosity (70%) piezoelectric device was designed using SolidWorks as a prototype of P-type TPMS structure. The independent unit of the model is a combination of a cube plus an octagonal tab with decreasing cross-sectional area extending in six directions, as shown in Figure 4. An optimized printing process was used to fabricate the complex piezoelectric device, and its potential applications were explored. Using a piezoelectric device with a porosity of 70% as an energy harvesting unit, it was applied in both force–electric sensing and magneto–electric sensing. Conductive silver wires were soldered to both sides of the piezoelectric device and connected to an oscilloscope to capture the voltage signal generated by the device. A crank rocker mechanism driven by an electric motor was used to execute a fixed-frequency reciprocating tapping action, and the signal generated by the piezoelectric element was recorded, as shown in Figure 5. The tip of the rocker arm for the force–electric sensing part was equipped with a soft latex material that could apply a standard force to the piezoelectric device while not damaging it. The position of the movable fixed platform was set to 0 displacement position when the tip just contacted the piezoelectric device at the end of the stroke. Next, the platform was moved 1 mm to the left to release a larger standard force, which was repeated five times. For the magneto–electric sensing part, the tip of the rocker’s arm was equipped with a magnet, and the platform positions of two like-poles that were about to make contact were defined as the 0 displacement position. Then, the platform was moved 1 mm to the right to release a smaller magnetic repulsion force, which was repeated five times. The waveform diagrams of the piezoelectric changes under different platform displacement conditions are shown in Figure 6. It was found that as the displacement increases, the piezoelectric device for force–electric sensing can generate a larger voltage, and the ratio of voltage increase is much higher than the ratio of displacement increase. However, the voltage generated by the piezoelectric device for magneto–electric sensing decreases with the increase in displacement. This experimental result proves the force–electric conversion and magneto–electric sensing ability of the piezoelectric device, and it can realize functions such as weight measurement, force signal monitoring, impact energy harvesting, and magnetic force capture to a certain extent, which lays a foundation for the manufacture of simple force sensors and magnetic sensors.

## 4. Conclusions

In this paper, the entire process of DLP printing lead-free piezoelectric ceramic BTO was optimized. When using a BTO powder with a particle size of 200 nm to prepare a ceramic slurry with a solid content of 75 wt%, adding a dispersant of 3% of the total mass (BTO powder + ceramic-based photosensitive resin) can result in a better printing efficiency and accuracy. In the heat treatment process, using a heating rate of 4 °C/min for degreasing, 4 °C/min for carbon removal, and 2 °C/min for sintering can obtain higher part density and integrity. During the poling process, a poling field of 10 kV/cm, a poling time of 50 min, and a poling temperature of 60 °C can achieve a higher piezoelectric constant (about 211 pC/N). We also designed and manufactured a high-porosity piezoelectric component and explored its applications in force–electric sensing and magneto–electric sensing. We optimized the entire printing process of BTO piezoelectric ceramics and achieved high-precision and high-performance piezoelectric ceramic manufacturing, which has significant implications for the development of electronic components. In addition, this printing process has many other potential applications, such as special complex piezoelectric devices that can promote uniform cell growth and can be applied to artificial bone direction. DLP printing technology can also be used to print composite devices with different shapes and component content to explore the related performance of piezoelectric devices with different composite methods.

## Figures and Tables

**Figure 1 micromachines-14-01146-f001:**
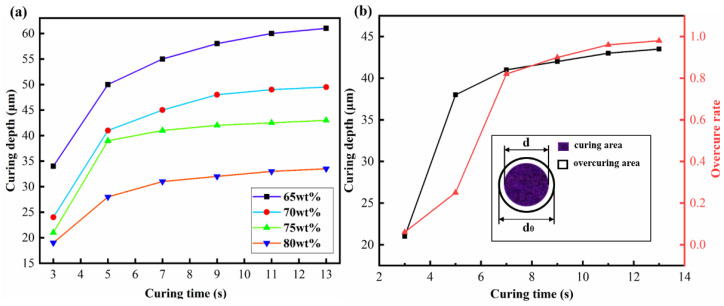
(**a**) The effect of powder-solid content on curing depth; (**b**) The effect of curing time on curing depth and overcuring rate. The schematic diagram for measuring the overcuring rate.

**Figure 2 micromachines-14-01146-f002:**
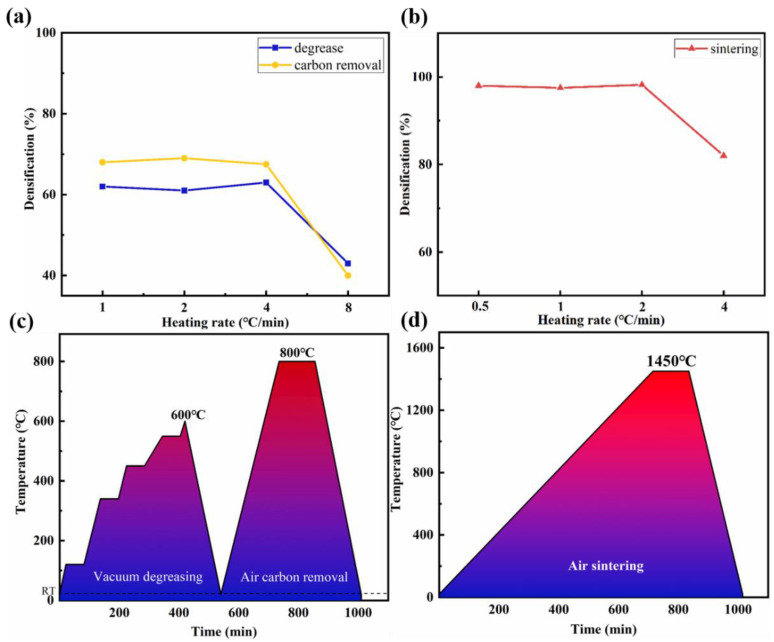
(**a**) The effect of temperature rise rate on the process of glue discharge; (**b**) the effect of temperature rise rate on the process of sintering; (**c**) temperature rise curve of the glue discharge process; (**d**) temperature rise curve of the sintering process.

**Figure 3 micromachines-14-01146-f003:**
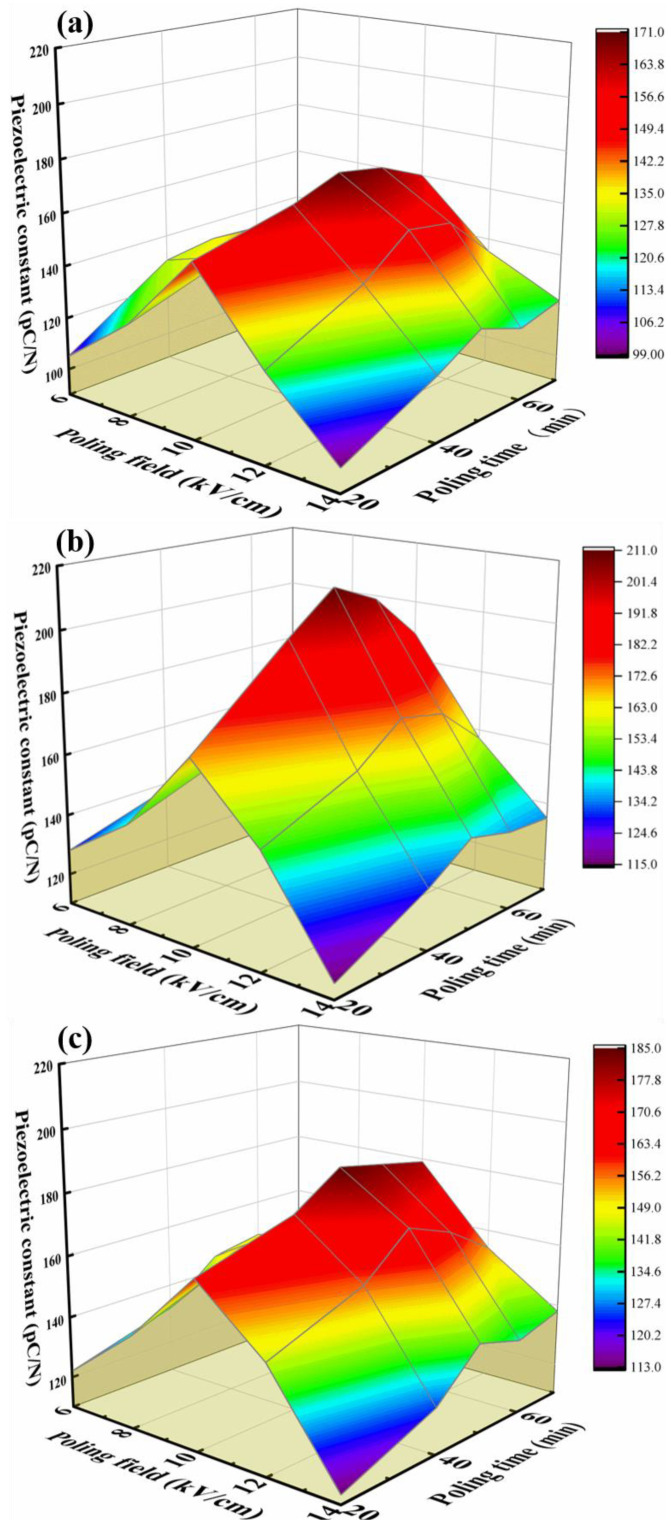
Effect of poling field and poling time on piezoelectric constant at different poling temperatures: (**a**) 40 °C; (**b**) 60 °C; (**c**) 80 °C.

**Figure 4 micromachines-14-01146-f004:**
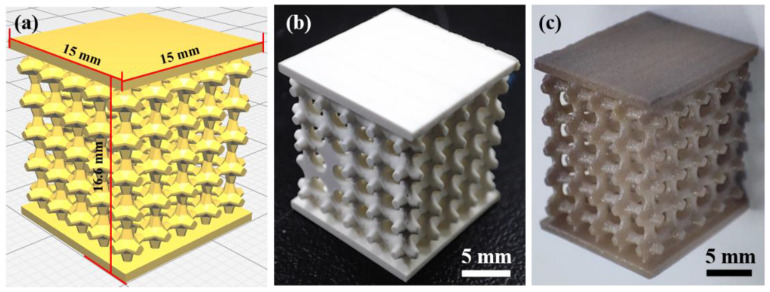
Model and physical view of the high porosity piezoelectric device: (**a**) printed model design drawing; (**b**) printed model green body; (**c**) printed parts after heat treatment.

**Figure 5 micromachines-14-01146-f005:**
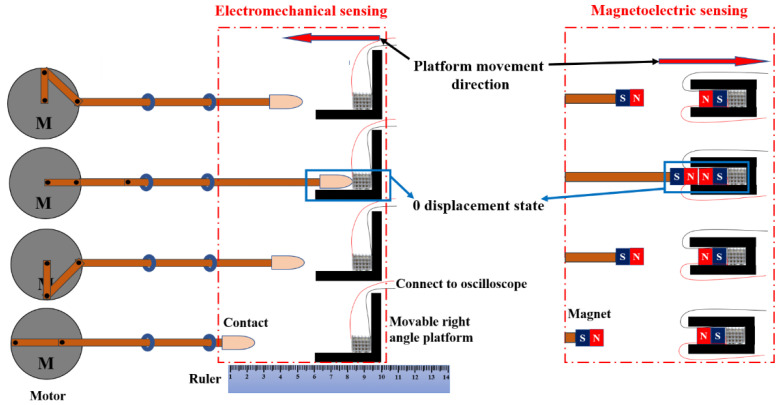
Schematic diagram of the crank rocker mechanism and diagram of the force and magnetoelectric sensing test setup.

**Figure 6 micromachines-14-01146-f006:**
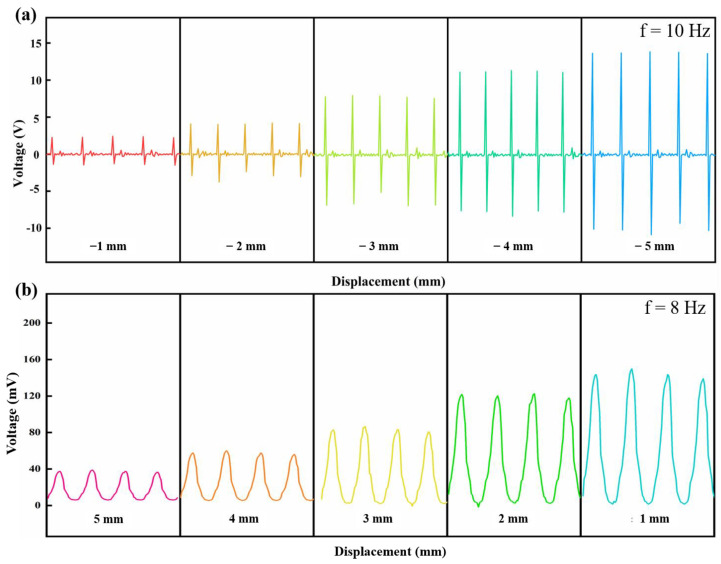
Waveform diagrams of voltage generated by piezoelectric devices under different platform displacements: (**a**) force electrical sensing waveform; (**b**) magnetoelectric sensing waveform.

**Table 1 micromachines-14-01146-t001:** Summary of optimized printing parameters.

Process Parameters	Optimized Values
Solid content	75 wt%
Curing time	5 s
Degreasing heating rate	4 °C/min
Carbon removal heating rate	4 °C/min
Sintering heating rate	2 °C/min
Poling field	10 kV/cm
Poling time	50 min
Poling temperature	60 °C

## Data Availability

All data needed to evaluate the conclusions in the paper are present in the paper.

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
