# Peer review of "Parameter Optimization for Printing Barium Titanate Piezoelectric Ceramics through Digital Light Processing"

_micromachines, 2023, doi:10.3390/mi14061146_

Round 1

Reviewer 1 Report

1.       It is better to add the reference number after "et al.".

2.       In the "Slurry preparation" section, how was the homogeneity of the slurry ensured?

3.       It is better to give the design of the experiment and the levels of the investigated parameters in a table.

4.       The use of a DLP printer is actually for printing complex geometries. Provide explanations about the geometry and application of the desired geometric shape. (The geometric shape shown in Figure 4).

Author Response

Point 1: It is better to add the reference number after "et al.".

Response 1: Thank you very much for pointing out these problems. We have made changes in the manuscript and marked them with special colours.

Point 2: In the "Slurry preparation" section, how was the homogeneity of the slurry ensured?

Response 2: We ensure the ability to suspend the powder particles in the slurry by adding the right amount of dispersant to the slurry in order to obtain a ceramic slurry with good homogeneity. Static experiments were conducted on slurry samples with different powder solid contents after preparation. The samples were allowed to settle for 1, 3, 5, and 7 days, respectively. The experiment revealed that all slurry samples had significant sedimentation, and the ceramic slurry still possessed curing ability even after being left stationary for 7 days.

Point 3: It is better to give the design of the experiment and the levels of the investigated parameters in a table.

Response 3: Thank you for pointing out the shortcomings in the manuscript. We summarize the optimized parameters in the whole process in Table 1 in Section 3.3.

Point 4: The use of a DLP printer is actually for printing complex geometries. Provide explanations about the geometry and application of the desired geometric shape. (The geometric shape shown in Figure 4).

Response 4: Thank you very much for your valuable feedback. The model is a piezoelectric device designed after the P-type TPMS structure, which has good mechanical properties, energy absorption energy and printable lines. We add a description of the shape of the model and the specific modeling approach in Section 3.4 as “Triply periodic minimal surface (TPMS) structure has the advantages of being lightweight,…….”.

Reviewer 2 Report

This paper focuses on the Digital Light Processing technology and presents a study aimed to investigate and optimize the manufacturing of barium titanate piezoelectric ceramics. The paper presents an experimental study of the effects of the slurry composition and curing time in the printing process, heating rate in the heat treatment process, and of the poling field, time and temperature in the poling process. Moreover, the paper shows an example of manufacturing of a complex piezoelectric device. The topic of this study is timely and of interest for the manufacturing community, but the paper has to be improved to be considered for publication in a journal.

The “Introduction” section does not present an extensive review and discussion of the state of the art on the paper topic, in particular on the effect of the printing parameters on the DLP printing of piezoelectric ceramics. Therefore, this section is not entirely suitable to prove the paper novelty with respect to the existing literature.

The description of materials and methods have to be enhanced since many essential details are missing, including: machine details (i.e. manufacturer and characteristics, such as printing volume, resolution, etc.), geometry and number of the printed specimens, printing parameters (e.g. layer thickness), materials and times of the debonding process, number of specimens analyzed in each test, geometry of the complex piezoelectric device. These details are necessary to understand the experimental results and their interpretation that you provided.

You might also add some pictures of the printing / testing equipment and the printed specimens.

The technical terminology is sometimes not accurate [what is a “doctor blade”?], thus a revision of the text is recommended.

Author Response

Point 1: The “Introduction” section does not present an extensive review and discussion of the state of the art on the paper topic, in particular on the effect of the printing parameters on the DLP printing of piezoelectric ceramics. Therefore, this section is not entirely suitable to prove the paper novelty with respect to the existing literature.

Response 1: Thank you very much for pointing out this issue. The study did not provide an in-depth discussion of printing parameters in the process, and as such, there was no discussion on the research status of this aspect in the introduction section. To avoid misleading readers, we removed the statement regarding printing parameters from the introduction. Furthermore, we expanded the discussion on our target research parameters. For instance, other researchers tend to focus on the viscosity effect in studies related to solid content, the impact of the highest sintering temperature in heat treatment research, and the polarization intensity effect in polarization process research. In this study, we provided additional insights into the impact of various factors affecting the printing outcome in DLP printing technology, complementing the existing research and enhancing its completeness and efficiency. We added related content in the manuscript and marked them with special colors-red for new content and yellow for modified content.

Point 2: The description of materials and methods have to be enhanced since many essential details are missing, including: machine details (i.e. manufacturer and characteristics, such as printing volume, resolution, etc.), geometry and number of the printed specimens, printing parameters (e.g. layer thickness), materials and times of the debonding process, number of specimens analyzed in each test, geometry of the complex piezoelectric device. These details are necessary to understand the experimental results and their interpretation that you provided.

Response 2: Thank you for your suggestion. We have added the manufacturer of the printing equipment and some device parameters in the characterization and testing section. For each studied variable, we conducted no less than three repeated experiments, and the obtained experimental values are all averages. We added model diagrams and physical pictures of the printed examples, and marked the key dimensions used in model design. The heating curves during the stripping and sintering process were summarized in Figure 2. All changes are clearly reflected in the manuscript and are marked in different colors, with red representing new content and yellow representing modified content.

Point 3: You might also add some pictures of the printing / testing equipment and the printed specimens.

Response 3: Thank you for your suggestion. We have added a printed model drawing and a physical drawing to the manuscript, as shown in Figure 4. We have added equipment information and manufacturer details in the manuscript for the graphics related to printing equipment. Related pictures of the equipment can be searched on the Internet, such as ceramic printing equipment, rheological property measurement devices, tube furnaces, and sintering furnaces.

Reviewer 3 Report

In this manuscript, Zhang et al. reports the optimization of key process parameters including slurry formulation, heat treatment, and poling process for DLP printing lead-free piezoelectric ceramic BTO. As a result, the optimized printing process yields a high-porosity piezoelectric device, and such device is validated in force-electric sensing and magneto-electric sensing applications. This paper is complete, logical, and well-written. The work can be considered for the publication in this journal after revising the following minor issues:

1. In the Abstract, “The resulting parts are polarized using a poling field of 10 V/cm,…”. It should be “10 kV/cm”.

2. In Page 7, “Comparing the piezoelectric constants of piezoelectric parts polarized at three different temperatures, it was found that the optimal poling temperature is 50 ℃.” It should be “60 ℃”.

Author Response

Point 1: In the Abstract, “The resulting parts are polarized using a poling field of 10 V/cm,…”. It should be “10 kV/cm”.

Response 1: Thank you for pointing out this typo. We have made changes in the manuscript and marked them in red.

Point 2: In Page 7, “Comparing the piezoelectric constants of piezoelectric parts polarized at three different temperatures, it was found that the optimal poling temperature is 50 ℃.” It should be “60 ℃”.

Response 2: Thank you for pointing out this mistake. We have made changes in the manuscript and marked them in red.

Reviewer 4 Report

This work optimizes the parameters of DLP printing of ceramic BTO. The optimized conditions were used to make a sensor device to validate the feasible applications. This study offers an inspiring avenue for DLP technology and is recommended to be accepted after addressing the following issues:

1.      The figure alphabetic number is recommended to put outside the figure frame.

2.      On page 4, the expression, ‘it is best to use BTO ceramic slurry…’ may be overtone. It can be revised as: ‘it is preferred…’ or other similar expression.

3.      Can the authors discuss more about the potential real-world applications of this technology?

The English expression should be minorly improved.

Author Response

Point 1: The figure alphabetic number is recommended to put outside the figure frame.

Response 1: Thank you for pointing out this typo. We have relocated the letter labels in all data figures.

Point 2: On page 4, the expression, ‘it is best to use BTO ceramic slurry…’ may be overtone. It can be revised as: ‘it is preferred…’ or other similar expression.

Response 2: Thank you for pointing out this mistake. We have accepted your suggestion and made corresponding modifications in the manuscript.

Point 3: Can the authors discuss more about the potential real-world applications of this technology?

Response 3: Thank you for pointing out this deficiency. We have added some potential applications of this printing process in the summary section of our manuscript.” In addition, This printing process has many other potential applications……”

Round 2

Reviewer 2 Report

The paper quality has been enhanced taking into account the comments and the paper can now be considered for publication.

[plain text = my previous comments, grey = answers by paper authors, blue = my new comments]

Point 2: The description of materials and methods have to be enhanced since many essential details are missing, including: machine details (i.e. manufacturer and characteristics, such as printing volume, resolution, etc.), geometry and number of the printed specimens, printing parameters (e.g. layer thickness), materials and times of the debonding process, number of specimens analyzed in each test, geometry of the complex piezoelectric device. These details are necessary to understand the experimental results and their interpretation that you provided.

Response 2: Thank you for your suggestion. We have added the manufacturer of the printing equipment and some device parameters in the characterization and testing section. For each studied variable, we conducted no less than three repeated experiments, and the obtained experimental values are all averages. We added model diagrams and physical pictures of the printed examples, and marked the key dimensions used in model design. The heating curves during the stripping and sintering process were summarized in Figure 2. All changes are clearly reflected in the manuscript and are marked in different colors, with red representing new content and yellow representing modified content.

The geometry of the simple specimens for the characterization tests is still not described in the paper.

The constant printing parameters should be introduced before Table 1 (which is nearly at the end of the “Results and discussion” section), possibly in the “Materials and methods” section.

The text of the X- and Y-axis of the graphs in Fig. 3 is difficult to read.

Point 3: You might also add some pictures of the printing / testing equipment and the printed specimens.

Response 3: Thank you for your suggestion. We have added a printed model drawing and a physical drawing to the manuscript, as shown in Figure 4. We have added equipment information and manufacturer details in the manuscript for the graphics related to printing equipment. Related pictures of the equipment can be searched on the Internet, such as ceramic printing equipment, rheological property measurement devices, tube furnaces, and sintering furnaces.

I am well aware that pictures of the equipment can be found on the Internet, but it would be interesting to see your actual samples in the printing and measurement set-up.

Minor editing of English language required

Author Response

Point 1: The geometry of the simple specimens for the characterization tests is still not described in the paper.

Response 1: Thank you for raising this question again and helping us improve this manuscript. We have added relevant details about the sample shapes for the over-curing rate test, heat treatment, and polarization processes. We added this information to the manuscript and highlighted it in red.

Point 2: The constant printing parameters should be introduced before Table 1 (which is nearly at the end of the “Results and discussion” section), possibly in the “Materials and methods” section.

Response 2: Thank you for your valuable suggestion. We have added specific printing parameters to Section 2.4 and replaced the previous table with textual descriptions. Please refer to the red markings in the manuscript for details.

Point 3: The text of the X- and Y-axis of the graphs in Fig. 3 is difficult to read.

Response 3: Thank you for your suggestion. We have redrawn Figure 3 and made modifications in the manuscript, which are highlighted in yellow. The axis information in the new figure is clearly visible.
